# The Characterization of a Subependymal Giant Astrocytoma-Like Cell Line from Murine Astrocyte with mTORC1 Hyperactivation

**DOI:** 10.3390/ijms22084116

**Published:** 2021-04-16

**Authors:** Xin Tang, Gabrielle Angst, Michael Haas, Fuchun Yang, Chenran Wang

**Affiliations:** Department of Cancer Biology, University of Cincinnati College Medicine, Cincinnati, OH 45267, USA; tangx5@ucmail.uc.edu (X.T.); angstgl@ucmail.uc.edu (G.A.); haasmk@ucmail.uc.edu (M.H.); yangfu@ucmail.uc.edu (F.Y.)

**Keywords:** TSC1, mTORC1, astrocyte, SEGA-like tumorigenesis, rapamycin

## Abstract

Tuberous sclerosis complex (TSC) is a genetic disorder caused by inactivating mutations in TSC1 (hamartin) or TSC2 (tuberin), crucial negative regulators of the mechanistic target of rapamycin complex 1 (mTORC1) signaling pathway. TSC affects multiple organs including the brain. The neurologic manifestation is characterized by cortical tubers, subependymal nodules (SEN), and subependymal giant cell astrocytoma (SEGA) in brain. SEGAs may result in hydrocephalus in TSC patients and mTORC1 inhibitors are the current recommended therapy for SEGA. Nevertheless, a major limitation in the research for SEGA is the lack of cell lines or animal models for mechanistic investigations and development of novel therapy. In this study, we generated TSC1-deficient neural cells from spontaneously immortalized mouse astrocytes in an attempt to mimic human SEGA. The TSC1-deficient cells exhibit mTORC1 hyperactivation and characteristics of transition from astrocytes to neural stem/progenitor cell phenotypes. Rapamycin efficiently decreased mTORC1 activity of these TSC1-deficient cells in vitro. In vivo, TSC1-deficient cells could form SEGA-like tumors and Rapamycin treatment decreased tumor growth. Collectively, our study generates a novel SEGA-like cell line that is invaluable for studying mTORC1-driven molecular and pathological alterations in neurologic tissue. These SEGA-like cells also provide opportunities for the development of novel therapeutic strategy for TSC patients with SEGA.

## 1. Introduction

Tuberous Sclerosis Complex (TSC) is an autosomal dominantly inherited neurocutaneous disorder caused by mutations in either TSC1 or TSC2 and characterized by development of benign tumors in multiple different organs [1]. More than 90% of TSC patients show the presence of three main types of brain lesions: cortical or subcortical tubers, subependymal nodules (SENs), and subependymal giant cell astrocytoma (SEGAs). SEGA is a low-grade brain tumor occurring in ~20% of TSC patients. In the majority of cases, SEGAs arise in the first two decades of life, from the head of the caudate nucleus near the foramen of Monro and grow inside the lateral ventricle [2]. SEGAs can cause life threatening complications due to hydrocephalus or due to intratumoral hemorrhage [3]. Currently, it is still not known how TSC-deficient neural cells eventually grow and become SEGAs.

Our understanding of SEGAs mainly stems from their typical histopathological descriptions: SEGAs are solid sheets and perivascular pseudorosettes of large, gemistocytic, polygonal and occasionally ganglion-like cells within a fibrillated background, accompanied by spindle-shaped cells creating broad fascicles [2]. At this point, many cell lines have been generated with loss of either TSC1 or TSC2, and these have been extremely valuable in understanding the effects of mTORC1 hyperactivation, but none have faithful pathological features of neural tissue or SEGA in TSC patients. This is a major limitation in SEGA research to understand how these tumors arise and to find novel therapeutics. Many labs interested in TSC tried to develop rodent models to study TSC-related brain lesions or SEGA. Zhou et al. used tamoxifen-inducible Nestin-CRE or Ascl1-CRE to conditionally knock out Tsc1 in neural stem/progenitor cells (NSCs) [4]. TSC1 deletion caused mTORC1 hyperactivation and small ventricular lesions reminiscent of SENs. However, well-defined SEGAs were not observed in these mouse models. Some mouse models have limited similarity to human SEGAs because of the short lifespan of animal with brain TSC deficiency [5]. Other mouse models have required additional genetic events, such as loss of PTEN, which does not occur in TSC patients [6]. In addition, genetically engineered human cortical spheroid models and neural cell model of TSC from embryonic stem cell (ES) or induced pluripotent stem cells (iPS) with loss/mutation of TSC1/TSC2 have been generated [7,8]. However, their similarity to human TSC and SEGA is uncertain, and engraftment of such cell lines in immunodeficient mice has not been achieved, which is a hallmark of a tumor cell line.

mTOR inhibitors (rapamycin or Rapalogs) are the main therapy for SEGA in TSC patients [9]. Franz et al. reported regression of SEGAs in TSC patients following rapamycin treatment [10]. New rapamycin analogs (Rapalogs) have been developed for TSC patients and other diseases with mTORC1 hyperactivation. Everolimus, the hydroxyethyl ester of rapamycin, was successful in shrinking SEGAs [11,12]. Long-term Everolimus treatment efficiently reduced SEGAs and prevented new SEGAs formation [13]. In addition, Everolimus has been used to decrease the severity of epilepsy and ameliorate neuropsychiatric problems in TSC patients [14,15]. The mTOR inhibitors can also be used for several rare neurodevelopmental disorders in which the hyperactivation of the mTORC1 pathway has been verified. However, prolonged treatment of SEGA is necessary, since tumors can regrow after Rapalog treatment is discontinued. Thus, there is an unmet need to find a platform to test novel therapeutic interventions for SEGA.

In this study, we have shown the spontaneous conversion of primary astrocyte into cells capable of growing into a defined SEGA-like tumor in the absence of widespread genomic changes. These cells should therefore be useful in identifying the factors involved in SEGA-like tumor initiation and in defining how each of the required changes contributes to tumorigenesis. This system will also be useful in identifying other therapeutic opportunities to improve SEGA treatment.

## 2. Results

### 2.1. Spontaneous Immortalization of Isolated Astrocytes from Tsc1^GFAP^ cKO Mouse

To facilitate our research of TSC-deficient neural cells, we isolated primary astrocyte from neonatal WT and Tsc1^GFAP^ cKO mice separately. These primary astrocytes were cultured in vitro for 355 days (primary culture established on 12 March 2018 and first passage on 2 March 2019) when we noticed that the media of two wells from different animals became yellow. We dissociated the two cells and cultured them individually. By the time we submitted the manuscript, both cell lines had been passaged for more than 200 times. We genotyped these cells and confirmed that one line is originated from a Tsc1 flox/fox WT mouse (designated as Ast #1) and the other is from a Tsc1^GFAP^ cKO mouse (designated as Ast #2). We did not observe an obvious difference in morphology between Ast #1 and Ast #2 (as shown in Figure 1A), but we noticed that Ast #1 cells spread faster than #2 cells after plating. Western blot data indicated a diminished TSC1 expression in Ast #2 compared to Ast #1 cells (Figure 1B), consistent with the genotype data. Using primary astrocytes as control, we also found decreased TSC1 expression in Tsc1^GFAP^ cKO astrocyte compared to WT astrocytes (Figure 1B). Consistent with the decreased TSC1 level, the TSC2 level also decreased in Ast #2 cells as TSC1 functions through stabilizing TSC2 (Figure 1C). As expected, mTORC1 activity significantly increased in Ast #2 cells as indicated by the elevated levels of pS6RP (phosphorylated S6 ribosomal protein) and p70 S6 kinase (pS6K, phosphorylated S6 ribosomal protein kinase) when they were compared to Ast #1 cells (Figure 1C). We treated both Ast #1 and Ast #2 cells with Rapamycin at 100 nM, which abolished the pS6RP and pS6K levels in these cells, suggesting the inhibition of mTORC1 activity (Figure 1C). TSC1 and TSC2 levels were not affected by Rapamycin treatment (Figure 1C). These results indicated that loss of TSC1 would cause mTORC1 hyperactivation in immortalized astrocyte.

Next, we analyzed the neural stem/progenitor cell (NSC) markers and astrocyte markers expressed in Ast #1 and Ast #2 around the passages of 75. The protein level of astrocytic maker GFAP (glial fibrillary acidic protein) and GS (glutamine synthetase) significantly decreased in both Ast #1 and #2 compared with primary WT astrocytes (Figure 1D). In contrast, there was a marked increase of Sox2 (SRY-Box Transcription Factor 2) protein level in Ast #1 and Ast #2 compared to primary astrocytes. There was also a modest increase of nestin level in the immortalized cell lines (Figure 1D). Immunofluorescent staining showed nuclei localization of Sox2 and cytoplasmic localization of intermediate filament nestin in both Ast #1 and Ast #2 cell lines (Figure 1E). We used qRT-PCR to check the NSC markers in Ast #1 and Ast #2 cells as well as in neurospheres from WT and Tsc1^GFAP^ cKO mice. Consistent with the immunofluorescent staining, we found comparable levels of *nestin* mRNA and *Sox2* mRNA expression in both Ast #1 and Ast #2 cells (Figure 1F). We noticed more than 3-folds decreased expression of *nestin* mRNA and *Sox2* mRNA in immortalized cell lines compared to those in neurospheres (Figure 1F). Next, we cultured the Ast #1 and Ast #2 cells under suspension conditions and we found that both cell lines formed spheres with similar size (Figure 1G). Serial dilution of seeding cells for spheres indicated that there was no difference in the sphere formation ability for both Ast #1 and Ast #2 cells (Figure 1G). With the gain of some NSC/progenitor characters, we noticed a dramatic decrease of astrocyte markers of *GFAP* and *Aqp4* (Aquaporin 4) in Ast #1 and Ast #2 cells. Their mRNA levels were only 1–2% (*GFAP*) or less than 1% (*Aqp4*) of those in primary astrocytes (Figure 1H). The mRNA level of another astrocyte marker, *Cx43*, was also significantly lower in Ast #1 and Ast #2 cells compared to primary astrocytes (Figure 1H). mRNA level of *Cx43*, but not *GFAP* and *Aqp4*, in Ast #2 cells was significantly higher than that in Ast #1 cells (Figure 1H). Taken together, these data suggested that the Ast #1 and Ast #2 cells obtained some NSC/progenitor status during the process of gaining immortalization.

### 2.2. Passaging of Immortalized TSC1-Deficient Astrocytes

We performed the proliferation assay at different passages (32, 64, 100, 162, 200 for Ast #1 and 32, 62, 98, 160, 198 for Ast #2) and examined cell numbers at different days post-seeding. Our results indicated that both cell lines could rapidly proliferate from 50,000 cells to 1.8 × 10^6^–2 × 10^6^ cells after 5 days in culture at different passages (Figure 2A). The number of Ast #2 cells was slightly higher than that of Ast #1 after 5 days in culture in passages of 32, 64, and 200 (Figure 2A). We stained proliferative marker of Ki67 at the passage of 200 and we found that both cell lines showed >90% Ki67^+^ cells one day after plating (Figure 2B,D). Rapamycin treatment significantly decreased their growth and one-day Rapamycin treatment already significantly decreased Ki67 positive cells in both Ast #1 cells and Ast #2 cells (Figure 2C,D). Interestingly, we noticed that Ast #2 cells proliferate faster and have more Ki67^+^ cells after Rapamycin treatment when compared to Ast #1 cells with same treatment (Figure 2C,D). These results suggested that the two cell lines had successfully passed the Hayflick limit and cellular senescence, which happens in normal cells around 50 passages in vitro. Indeed, our β-gal staining did not reveal positive staining of Ast #1 and Ast #2 cells at passage 35 (Figure 3A). Using old primary astrocyte (>12 month in culture) as a positive control, we found that ~70% of old astrocytes were β-gal positive (Figure 3A). Because p53 pathway is important for cell cycle arrest and cellular senescence [16], next, we performed qRT-PCR to examine the *p53* and its target *p21* in immortalized cells as well as in primary WT and TSC1 cKO neurospheres. We found that both Ast #1 and Ast #2 cells had significantly lower mRNA levels of *p53* and *p21* compared to those in primary neurospheres (Figure 3B). Deletion of TSC1 increased *p21* mRNA level in Ast #2 cells but not in TSC1-null neurospheres (Figure 3B,C). Rapamycin treatment did not affect the expression of *p53* mRNA in both cell lines, while it decreased *p21* mRNA level in Ast #2 cells (Figure 3C). Even overactive mTORC1 usually enhances the activation of *p53* to drive cells to senescence [17]; however, the downregulation of *p53* gene expression in both TSC1-competent Ast #1 and TSC1-null Ast #2 cells suggested an unexplored mTORC1-activity independent mechanism to overcome senescence.

### 2.3. Tumorigenesis of Immortalized TSC1-Deficient Astrocyte to form SEGA-Like Tumors

To examine the tumorigenesis of the immortalized cells, we transplanted Ast #1 and Ast #2 cells into the nude mice for subcutaneous (S.C.) growth. We found that the Ast #1 tumors grew slower than Ast #2 tumors (Figure 4A,B). Seven days after transplantation, we treated the mice bearing Ast #2 tumors with vehicle or Rapamycin. We found that Rapamycin significantly reduced the growth and weight of tumors (Figure 4A,B).

For histology of the Ast #2 tumors, we used the SEN-like structure from brains of Tsc1^GFAP^ cKO mice [5] as a control. The cells in the SEN-like structure were small with hyperchromatic nuclei appearance (Figure 4C-c1). We found that the tumors from Ast #2 cells were intermixed with at least three major distinct cell populations. We noticed some small cells with hyperchromatic nuclei and glial appearance (arrows in Figure 4C-c2), similar to what we observed in the SEN-like structure (Figure 4C-c1). There were normal-sized cells with a spindle shape with long processes intersecting (arrowheads in Figure 4C-c3). Another type was cells with balloon-like morphology. These cells had very large soma with eosinophilic glassy cytoplasm (arrows in Figure 4C-c3,c4). We also found scarcely represented multinucleated giant cells in tumors from Ast #2 cells (arrowhead in Figure 4C-c4). Rapamycin-treated tumors were characterized by low cellularity and by spindle-shaped cells with large eosinophilic cytoplasm and round or oval nuclei (arrows in Figure 4C-c5). Ast #1 tumors showed similar histology as Rapamycin-treated Ast #2 tumors (Figure 4C-c6). Next, we examined the activation of mTORC1 by staining of pS6. We found modest-to-low pS6 staining in Ast #1 tumors and high pS6 signals in the majority of parts of tumors from Ast #2 cells (Figure 4D). Rapamycin treatment significantly reduced the extent of pS6 signaling in Ast #2 tumors (Figure 4D). These results indicated that tumors derived from Ast #2 cells exhibited some SEGA-like phenotypes in vivo and Rapamycin could reduce the tumor growth and some SEGA characters through suppressing mTORC1 activity.

### 2.4. mTORC1-Dependent Survival and Differentiation of SEGA-Like Ast #2 Tumors

Next, we examined the proliferation of tumors from Ast #2 and we found 20 ± 5.5% Ki67^+^ cells. Rapamycin treatment had no effect on the proliferation of Ast #2 tumor cells (22 ± 3.6%, *p* = 0.65) (Figure 5A). Instead, Rapamycin significantly increased the number of TUNEL^+^ cells (2.8 ± 1.5% in the vehicle group vs. 5.2 ± 1.3% in the Rapamycin group, *p* < 0.05) (Figure 5B), which might account for the reduced tumor size (Figure 4B). Then, we analyzed the differentiation of Ast #2 tumor cells with or without Rapamycin treatment. We found scattered doublecortin positive (DCX^+^) cells (arrow in Figure 5C) in Ast #2 tumors. The percentage of DCX^+^ cells in Ast2 tumor was ~1% of total cells. Rapamycin treatment abolished the generation of DCX^+^ cells in Ast #2 tumors (Figure 5C), suggesting that high mTORC1 activity drive their neuronal differentiation. We found 10% GFAP^+^ cells in the TSC1-null tumors, and Rapamycin treatment had no effect on the astrocytic differentiation (Figure 5D). Similarly, Rapamycin treatment had no effect on the Sox2 expression (Figure 5E). We did not detect any NeuN^+^ cells in Ast #2 tumor irrespective of Rapamycin treatment (data not shown). Together, these data indicated that mTORC1 hyperactivation prevented cell death in Ast #2 tumors and drove their differentiation with mixed neuron-glia lineages.

### 2.5. Restoration of mTORC1 Activation in SEGA-Like Tumors by Replenish Human TSC1

To verify the consequence of TSC1 loss in SEGA-like tumors, next, we used recombinant lentivirus encoding human TSC1 (hTSC1) to infect the TSC1-deficienct cells. Western blot indicated that hTSC1 lentivirus restored the TSC1 level in Ast #2 cells (Figure 6A). The highly activated mTORC1 in Ast #2 cells, as indicated by phosphorylated S6RP, was rescued in Ast #2 + hTSC1 cells to the level as that in Ast #1 cells (Figure 6A). We found that Ast #2 + hTSC1 grew slower than Ast #2 cells infected with empty vector (Figure 6B). These Ast #2 + hTSC1 cells grew similar as Ast #1 cells after 5 days in culture (Figure 6B). Interestingly, Rapamycin significantly suppressed the growth of Ast #2 + hTSC1 cells (Figure 6B), suggesting that re-expression of TSC1 restored the proliferation and response of Ast #2 cells to Rapamycin.

We injected Ast #2 cells and Ast #2 + hTSC1 cells S.C. into nude mice and treated the tumors with vehicle or with Rapamycin. Ast #2 + hTSC1 cells grew smaller tumors than Ast #2 cells infected with empty vector (Figure 6C). The naïve Ast #2 + hTSC1 tumor was similar to Rapamycin-treated Ast #2 cells while Rapamycin further decreased the weight of Ast #2 + hTSC1 tumors (Figure 6C). The Ast #2 + hTSC1 tumors had lower pS6RP staining and Rapamycin treatment further reduced the pS6RP level to background level in vivo (Figure 6D). hTSC1 had little effect on the proliferation of Ast #2 + hTSC1 tumor cells, while Rapamycin treatment reduced the Ki67^+^ cells in these tumors (6.5 ± 2.3% with Rapamycin vs. 22.8 ± 3.5% with Vehicle, *p* < 0.01) (Figure 6E). Restoring hTSC1 in Ast #2 showed no difference in the percentage of TUNEL^+^ cells and Sox2^+^ cells in Ast #2 tumors (Figure 6F,G). Rapamycin had no effect on apoptosis (2.3 ± 0.9%with Rapamycin vs. 2.5 ± 1.1% with vehicle, *p* = 0.85) and Sox2^+^ cells in Ast #2 + hTSC1 tumors (Figure 6F,G). These results further indicated the functions of TSC1 in progression of SEGA-like tumors.

## 3. Discussion

The interests to study tissue specific phenotypes of TSC are constantly increasing. To date, molecular and biochemical studies on TSC have been performed typically by exploiting Tsc1- or Tsc2-null mouse embryonic fibroblasts or other cell lines. Although these cell lines are very efficient as in vitro models, they do not recapitulate tissue-specific features. This intensified need for basic and clinical research can be partially met by the generation and use of new tissue specific transgenic animal models [18]. Very recently, neural cells and cerebral organoids derived from iPS have been isolated from TSC patient specimens and these cellular models could reproduce some neurological defects seen in TSC [8,19,20]. However, there is still a need for brain-specific in vitro TSC models which could be easily handled and are affordable for most research labs. In the present manuscript, we characterized a spontaneously immortalized mouse cell line from Tsc1-null astrocytes with the potential as an in vitro model to study some neural-specific TSC phenotypes. Particular emphasis was placed on this new cell line’s response to Rapamycin to suppress mTORC1 hyperactivation and its ability to generate SEGA-like tumors in vivo.

TSC1-deficient brain tumors were intermixed with distinct cell populations, e.g., small cells with hyperchromatic nuclei and glial appearance, normal-sized cells with a spindle shape and long processes intersecting, and cells with balloon-like morphology. We found scarcely represented multinucleated giant cells (Figure 4C) in TSC1-null tumors, which are hallmarks of human SEGAs [21]. We observed Sox2^+^ neural progenitors, some differentiated GFAP^+^ glia-like cells and DCX^+^ immature neurons in TSC1-null tumors; however, we did not find mature neurons in these tumors which were reported in human SEGA samples [22]. The absence of NeuN^+^ cells might be a consequence of hyperactivated mTORC1 to block the neuronal maturation [23]. In addition, these TSC1-deficient tumors showed response to Rapamycin treatment. Instead of becoming cytostatic, Rapamycin treatment increased apoptosis in TSC1-deficient tumors (Figure 5) and the Rapamycin-treated tumor cells were spindle-shape with round or oval nuclei (Figure 4C). Rapamycin treatment eliminated DCX^+^ immature neurons in Ast #2 tumors but it did not show significant effect on the expression of Sox2 and GFAP (Figure 5). These data suggested the application of Ast #2 cells to generate in vivo preclinical model with some human SEGA manifestations.

The low mRNA and protein levels of GFAP (Figure 1) and the absence of GFAP staining (data not shown) in the immortalized cells might raise concerns about their astrocytic origin. It has previously been reported that GFAP expression depends highly on the mitotic status of cells. Accordingly, GFAP was downregulated in a proliferative astrocyte cell line [24]. This might explain the low abundance of *GFAP* mRNA in our immortalized cell lines which are proliferative. It is also becoming increasingly clear that a large subpopulation of astrocytes does not express GFAP, even under in vivo conditions [25]. Besides GFAP, our immortalized cell lines also lacked some other genuine astrocyte markers such as glutamine synthase, Aqp4, and Cx43. Consistent with the in vitro findings, we observed that 5–10% of GFAP^+^ cells differentiated from TSC1-deficient tumors (Figure 5). It has been noticed that TSC SEGAs contain variable amounts of GFAP [22] and previous studies also reported low GFAP abundance in their samples [26] and giant cells were not frequently GFAP^+^ in human SEGAs [27]. Along with the obtaining of NSC markers and differentiation ability in vivo, our data suggest that the future application of Ast #2 explores the mechanisms of NSC/SEGA but not for astrocyte research.

We also noticed a high proliferative index in TSC1-deficient tumors as shown by ~25% Ki67-positive cells in our samples (Figure 5A). This ratio of proliferation is higher than that in previous reports using human SEGA samples, which was usually less than 5% in the most active area [28]. Our previous observation found 3–5% Ki67^+^ cells in SEN-like structure in Tsc1^GFAP^ cKO mice [5]. Interestingly, there was a report of human SEGAs with atypical histological features mimicking malignant gliomas. These SEGAs exhibited distinct anaplastic features with high Ki67 labelling index in the range of 15–20% [29]. Evidence has not yet linked the loss of a TSC gene and a second non-TSC gene in these SEGAs. For our experiments, we used the cells between 180 and 200 passages for tumorigenesis and we did not notice a dramatic change of cell proliferation during our serial passages from 30 to 200 (Figure 2A). We do not think that using the earlier passage of the Ast #2 cells will decrease the proliferative index of TSC1-null tumors as these immortalized cells already passed senescence at their early passages (Figure 3) or even earlier if we examined. We found a low mRNA expression of p53 in immortalized cells, which might help them overcome senescence (Figure 3) and induce higher proliferation of TSC1-null tumors in vivo. However, whether there is correlation of high proliferation with low p53 expression in TSC1-null tumors and the mechanisms to regulate p53 expression in these immortalized cells still needs further investigation.

With the progress in gene therapy, people begin to test the possibility to use Adeno Associate Virus (AAV) to treat neurological diseases in preclinical models [30]. Recently, a mouse model of TSC2 was generated by AAV-Cre recombinase disruption of Tsc2-floxed alleles at birth with a shortened lifespan and brain pathology consistent with TSC phenotype. When these mice were injected intravenously with AAV9 expressing condensed TSC2 (cTSC2), the mean survival was extended with reduction in brain pathology [31]. Nevertheless, this study does not test whether the effect of AAV-cTSC2 could be boosted by treatment with low dose Rapamycin or other Rapalogs. Our data indicated that restoring human TSC1 by lentivirus in Ast #2 cells reduced mTORC1 activation (Figure 6). Although we are not pursuing a gene therapy in the current manuscript, we found that TSC1 replenishment along with Rapamycin further decreased proliferation and growth in TSC1-deficient tumors (Figure 6C–E), suggesting an effective strategy for TSC patients with the combinational therapy.

In summary, the spontaneously immortalized mouse astrocytes described in our study provide a powerful model that uniquely mimics the cellular and molecular features of TSC brain lesion. Together with the corresponding tumor model, this cell line might be instrumental to identify lesion-specific molecular alterations and to test novel therapeutic vulnerabilities for TSC patients.

## 4. Materials and Methods

### 4.1. Animals

WT mice and Tsc1^GFAP^ cKO mice were described as before [5]. Eight-week-old nude mice (Jackson Laboratory, Bar Harbor, ME, USA) were used for analysis of tumor progression after transplantation. Mice were housed and handled according to local, state, and federal regulations. All experimental procedures were carried out according to the guidelines of Institutional Animal Care and Use Committee (IACUC) at University of Cincinnati (#18-06-21-2, approved on 20 October 2018).

### 4.2. Cell Cultures

Primary astrocytes were cultured as described previously [32]. Newborn pups were decapitated, and the cortices were removed from meninges, hippocampi, basal ganglia, and midbrain and kept in cold DMEM (Invitrogen, Carlsbad, CA, USA). The cortical tissue was cut into ~1 mm^3^ cubes and vortexed at the highest speed for 90 s. After filtration with 70 µm and 40 µm mesh, the cell suspension was plated in DMEM with 10% FBS onto tissue culture dishes (37 °C and 95% humidity). Medium was changed 3 d after plating, and FBS concentration was decreased to 7% 2 wks after seeding. For long-term astrocyte culture, we changed media twice every week till the cells were collected for analysis.

Neurosphere culture was performed as previously described [33,34,35]. In brief, SVZ tissue was isolated under a dissection microscope and cut into ~1 mm^3^ cubes. Tissue was digested in 0.2% trypsin/EDTA to obtain single-cell suspensions. Cells were cultured in neural basal media supplemented with B27 (Invitrogen), 10 ng/mL basic FGF, and 20 ng/mL EGF (Invitrogen) in Ultra-Low Attachment Plates (Corning Inc., Corning, NY, USA). Neurospheres with diameters larger than 50 µm were collected 10 d after culturing for RNA isolation.

Immortalized Ast #1 cells and Ast #2 cells were cultured in DMEM with 10% FBS under growth conditions. For serial passages, the cells were dissociated by 0.2% trypsin/EDTA (Invitrogen) solution every two to three days. In sphere culture, Ast #1 cells and Ast #2 cells with different number were cultured in neural basal media supplemented with B27 (Invitrogen), 10 ng/mL basic FGF, and 20 ng/mL EGF (Invitrogen) in Ultra-Low Attachment Plates (Corning) for 10 days.

### 4.3. Virus Production and Infection

Human Tsc1 cDNA encoded in pGIPZ lentiviral vector were generated from Open Biosystem (GE Healthcare Dharmacon, Lafayette, CO, USA). Package vectors of pSPAX2 and pMD2.G were mixed with pGIPZ vector and transfected in HK293 cells for virus production, essentially as described before [36]. Virus were concentrated by centrifuge and cells were infected and then incubated with Puromycin for selection.

### 4.4. Cell Proliferation Assay

Cells were seeded in 24-well tissue culture plates (5.0 × 10^4^ cells per well). After 2, 3, 4, and 5 days’ incubation, cells were digested by trypsin and resuspend. Cell number was counted in the indicated days with Countessmy of Life Science Technology (Thermo Scientific, Waltham, MA, USA). For passages of 200, the Rapamycin (100 nM, LC laboratory, Woburn, MA, USA) or vehicle was added into the culture medium for 5 days.

### 4.5. Subcutaneous Xenograft Models

Six nude mice were subcutaneously injected into the lateral backside with 1 × 10^6^ cells at their 180 passages. Cells were implanted into the right and left sides of each mouse. Calipers were used to measure the tumor volume, which was calculated using the formula Volume (π/6) × length × width^2^. For Rapamycin experiment, when tumor volume reached 100 mm^3^, mice were intraperitoneally injected with Rapamycin (5 mg/kg) every other day with 5% Tween 80 plus 5% PEG400 as vehicle. Mice treated with Rapamycin did not show any sign of toxicity based on body weight, food/water intake, and activity. We measured the tumor volume twice per week for 3–4 weeks before we collected the tumors to measure their weight. When mice were euthanized, their tumors were removed and sectioned with hematoxylin and eosin (H&E) staining and immunostaining.

### 4.6. Immunofluorescence

A total of 1 × 10^4^ cells was seeded into an 8-well chamber for 24 h. Then, cells were permeabilized with pure methanol for 10 min on ice, followed with 1% bovine serum albumin blocking buffer for 1 h at room temperature. Anti-Sox2, Ki67, and Nestin antibodies were incubated with cells in the blocking buffer overnight at 4 °C. Next, Alexa fluorescence 594 goat anti-rabbit was used for 1 h at room temperature, followed by ProLong^®^ Gold Antifade Mountant (Thermo Scientific). Images were acquired with Zeiss LSM confocal 710. Ki67^+^ cells were counted using Image J1.52i (Wayne Rasband; NIH, MD, USA).

### 4.7. Histology and Immunostaining

Mouse tumor tissues were kept in 4% paraformaldehyde at 4 °C. The specimens were then processed, embedded in paraffin and sectioned at 5 μm. Heat-induced antigen retrieval was performed in citrate buffer using a pressure cooker. Sections were incubated with primary antibodies overnight at 4 °C. Primary antibodies used were phosphorylated S6 ribosomal protein (Cell Signaling Technology, Danvers, MA, USA), S6 ribosomal protein (Cell Signaling Technology), Ki67 (Spring Bioscience Corporation, Pleasanton, CA, USA), GFAP (Cell Signaling Technology), Sox2 (Millipore Sigma, Burlington, MA, USA), DCX (Millipore). Sections were then incubated with FITC- or TRICT-conjugated IgG (for 1 h at room temperature), followed by DAPI staining (Vector Laboratories, Burlingame, CA, USA). Images were acquired with Olympus BX41 microscope. Six random fields from three to five different tissues in each group were quantified using ImageJ software.

### 4.8. Protein Extraction, SDS-PAGE and Western Blotting

Cells were collected for protein extraction with homogenization in modified radioimmune precipitation assay buffer (50 mM Tris-HCl, pH 7.4, 1% Triton X-100, 0.2% sodium deoxycholate, 0.2% SDS, 1 mM sodium EDTA) supplemented with protease inhibitors (5 μg/mL leupeptin, 5 μg/mL aprotinin, and 1 mM phenylmethylsulfonyl fluoride). After removing cell debris by centrifugation at 13,000 rpm for 10 min at 4 °C, protein concentration was determined using Bio-Rad protein assay reagent. The lysates were boiled for 5 min in 1 × SDS sample buffer (50 mM Tris-HCl, pH 6.8, 12.5% glycerol, 1% SDS, 0.01% bromphenol blue) containing 5% β-mercaptoethanol. They were then analyzed by SDS-PAGE followed by Western blot using various antibodies as described previously [17,18,19].

### 4.9. Real-Time PCR

Total RNAs were isolated from cells with GeneJET RNA Purification Kit (#K0731, Thermo Scientific) according to the user manual. Reverse transcription complementary DNAs (cDNAs) were synthesized with iScript cDNA Synthesis Kit (#1708891, Bio-Rad, Hercules, CA, USA). Real-time PCR was performed with iQ SYBR Green Supermix Kit (#170-8880, Bio-Rad). Expression values were normalized to β-actin. The primers were obtained from PrimerBank (https://pga.mgh.harvard.edu/primerbank/ (accessed on 21 January 2020)) unless specific references were cited. The specificity of all primers was validated with their dissociation curves.

### 4.10. Statistical Analysis

Statistical significance was evaluated by Two-way ANOVA, Student’s *t*-test, with *p* < 0.05 as indicative of statistical significance using Graph Pad Prism (Version 5.0, San Diego, CA, USA). The number of experiments used for quantification was indicated in the figure legends.

## Figures and Tables

**Figure 1 ijms-22-04116-f001:**
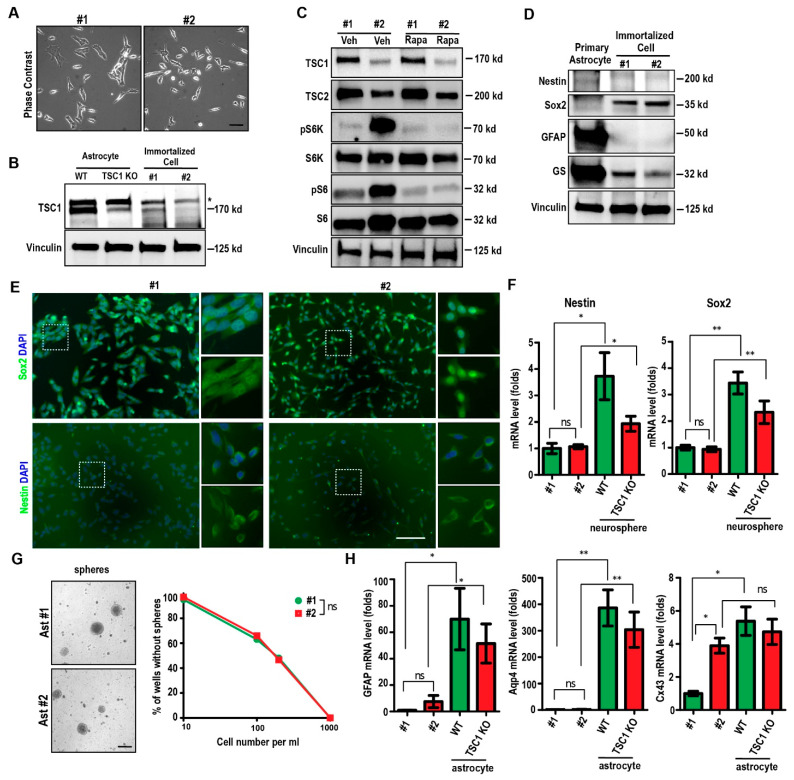
Characterization of immortalized cell lines Ast #1 and Ast #2 in vitro. (**A**) Phase contrast images of Ast #1 and Ast #2 cells after seeding for 1 day in culture. (**B**) Lysates extracted from primary astrocytes of WT mice and Tsc1 KO mice and immortalized astrocytic cell lines (#1 and #2) were analyzed by Western blot using antibodies for Tsc1 (top) and Vinculin (bottom). The star indicates a non-specific band revealed by Tsc1 antibody. Images were representative for 3 independent experiments. (**C**) Lysates from Ast #1 and Ast #2 treated with vehicle or Rapamycin for 24 h were analyzed by Western blot using antibodies for TSC1, TSC2, pS6K, total S6K, pS6RP, total S6RP, and Vinculin. Images were representative for 3 independent experiments. (**D**) Lysates extracted from primary WT astrocytes and immortalized cell lines were analyzed by Western blot using antibodies against GFAP, GS, Nestin, Sox2, and Vinculin as indicated. Images were representative for 3 independent experiments. (**E**) Immunofluorescent staining of Sox2 (upper panels), nestin (bottom panels), and DAPI in immortalized astrocytic cell lines. Boxed areas were shown in details on the right. (**F**) Mean ± S.E. of the relative mRNA expression levels of *nestin* and *Sox2* from Ast #1, Ast #2, and neurospheres from WT mice and Tsc1 cKO mice were shown. Four independent experiments were performed with similar results. (**G**) Phase contrast images of sphere formation from Ast #1 and Ast #2 cells. The sphere formation efficiency from serial diluted Ast #1 and Ast #2 at number of 10 cells per mL, 100 cells per mL, 200 cells per mL, and 1000 cells per mL are shown. Three independent experiments were performed with similar results. (**H**) Mean ± S.E. of the relative mRNA expression levels of *GFAP*, *Aqp4*, and *Cx43* from Ast #1, Ast #2 and astrocytes from WT and Tsc1 cKO mice were shown. Four independent experiments were performed with similar results. Two-way ANOVA and Tukey post hoc tests were used for statistic tests. ns: no significance; * *p* < 0.05, ** *p* < 0.01. Bar = 30 µm.

**Figure 2 ijms-22-04116-f002:**
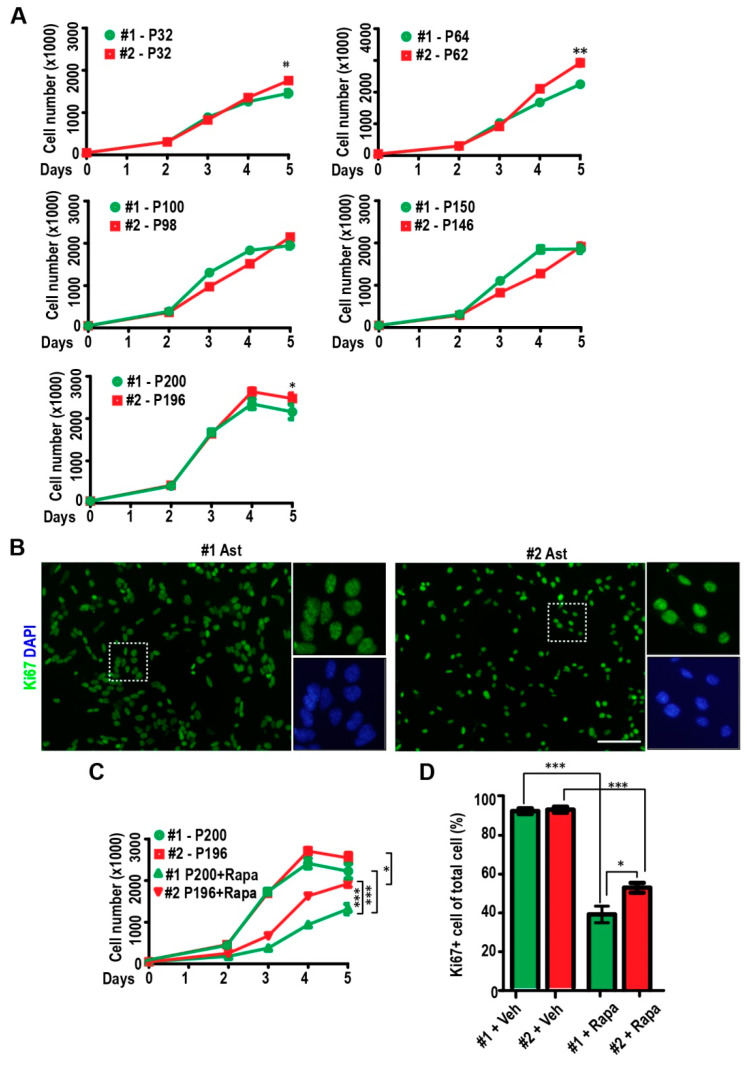
Growth of immortalized Ast #1 and Ast #2 cells in vitro. (**A**) Mean ± S.E. of cell number from Ast #1 and Ast #2 for 0, 2, 3, 4, and 5 days in culture at different passages were shown. Experiments were repeated 5–6 times for each time point. (**B**) Immunofluorescent staining of Ki67 and DAPI in immortalized astrocytic cell lines. Boxed areas are shown in detail on the right. Images are representative for 5 independent experiments. (**C**) Mean ± S.E. of cell number from Ast #1 and Ast #2 treated with vehicle or Rapamycin for 0, 2, 3, 4, and 5 days in culture at passages of 200 (Ast #1) and 196 (Ast #2) are shown. Experiments were repeated 5–6 times for each time point. (**D**) Mean ± S.E. of the percentage of Ki67 positive cells in Ast #1 and Ast #2 treated with vehicle or Rapamycin for 1 day in culture are shown. Experiments were repeated 5 times. More than 1000 cells were counted. Two-way ANOVA and Tukey post hoc tests were used for statistic tests. *: *p* < 0.05, **: *p* < 0.01, ***: *p* < 0.001. Bar = 50 µm.

**Figure 3 ijms-22-04116-f003:**
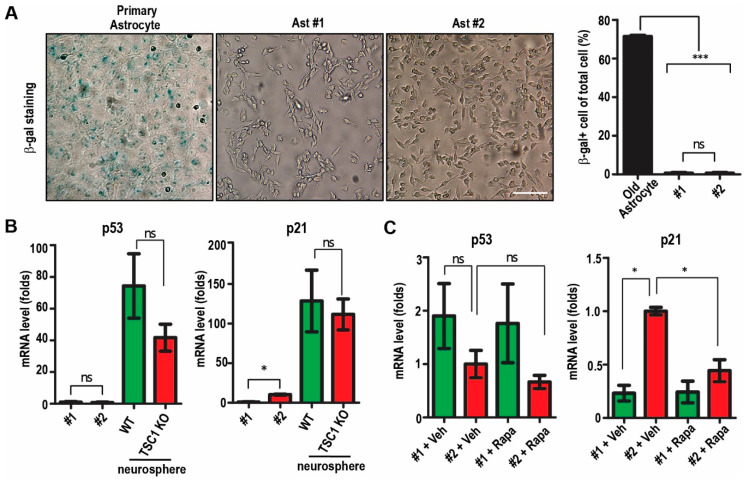
Immortalized cell lines of Ast #1 and Ast #2 passed senescence at early stage in vitro. (**A**) Representative images of β-gal staining for senescence from aged primary astrocyte (>1 year) and from immortalized Ast #1 and Ast #2 at passages of 35. Mean ± S.E. of the percentage of β-gal positive cells in aged astrocyte, Ast #1 and Ast #2 are shown on the right. Experiments were repeated 5 times for each type of cells. (**B**) Mean ± S.E. of the relative mRNA expression levels of *p53* and *p21* from Ast #1, Ast #2 and neurospheres from WT and Tsc1 cKO mice are shown. Four independent experiments were performed with similar results. (**C**) Mean ± S.E. of the relative mRNA expression levels of *p53* and *p21* from Ast #1, Ast #2 with or without Rapamycin treatment were shown. Four independent experiments were performed with similar results. Two-way ANOVA and Tukey post hoc tests were used for statistic tests. ns: no significance, * *p* < 0.05, ***: *p* < 0.001. Bar = 50 µm.

**Figure 4 ijms-22-04116-f004:**
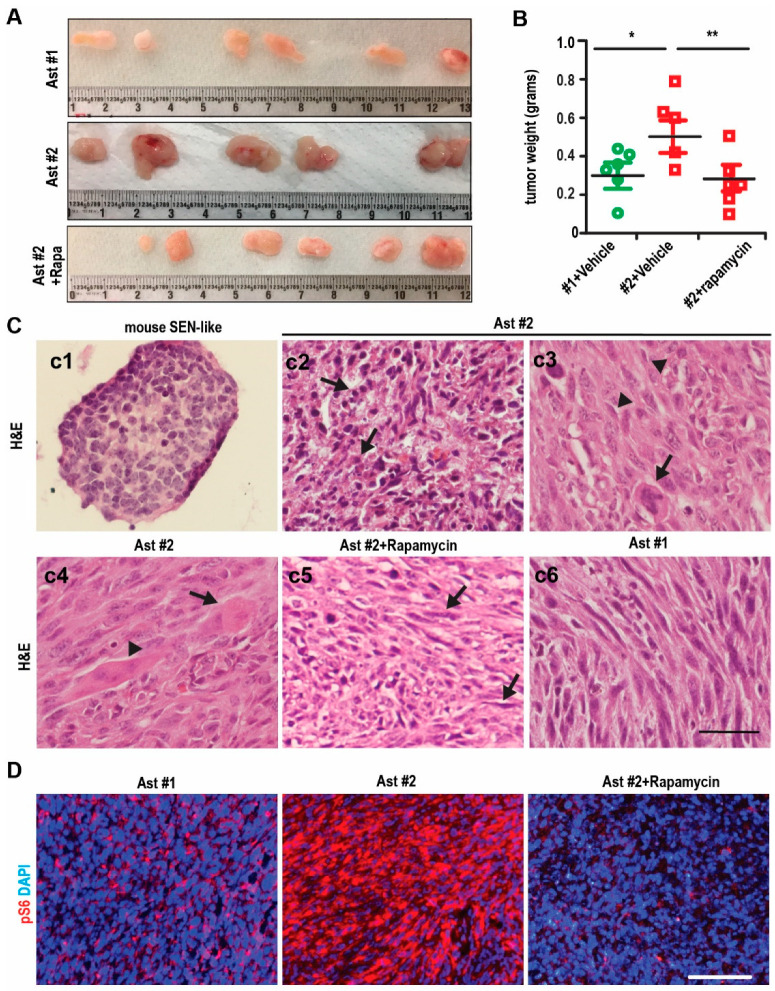
Formation of tumors from immortalized cell lines Ast #2 in vivo. (**A**) Representative images of tumors from immortalized Ast #1 cells treated with vehicle and Ast #2 cells treated with vehicle or Rapamycin for 2 weeks. (**B**) Mean ± S.E. of the tumor weight from Ast #1 cells treated with vehicle and Ast #2 cell treated with vehicle or Rapamycin for 2 weeks were shown. n = 6. (**C**) H&E staining of tumors from immortalized Ast #2 cells treated with vehicle or Rapamycin and Ast #1 cells treated with vehicle. (**D**) Immunofluorescent staining of pS6 and DAPI in Ast #1 tumors treated with vehicle or Ast #2 tumors treated with vehicle or Rapamycin. Two-way ANOVA and Tukey post hoc tests were used for statistic tests. * *p* < 0.05, ** *p* < 0.01. Bar = 50 µm.

**Figure 5 ijms-22-04116-f005:**
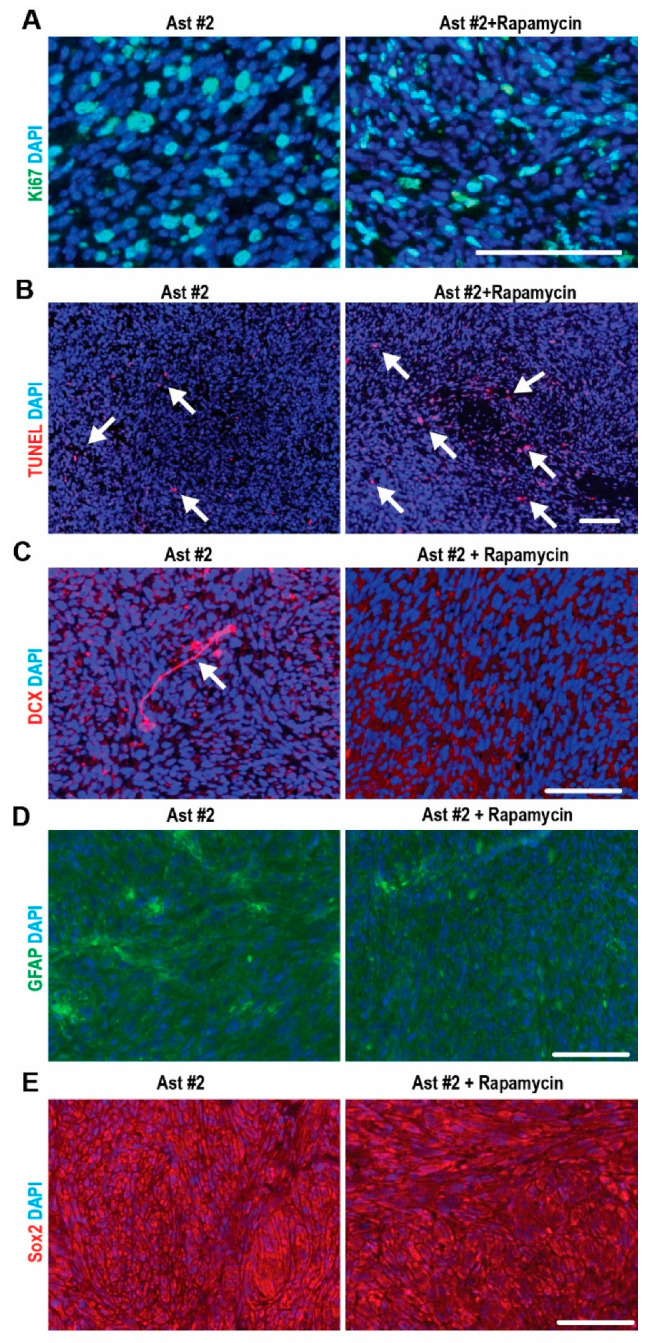
mTORC1-dependent and -independent differentiation in Ast #2 tumors. (**A**–**E**) Immunofluorescent staining Ki67 (**A**), TUNEL (**B**), DCX (**C**) GFAP (**D**), Sox2 (**E**) and DAPI of Ast #2 tumors treated with vehicle or Rapamycin for 2 weeks. Arrows in B indicated TUNEL-positive cells and arrows in C indicated DCX-positive cells. Bar = 50 µm.

**Figure 6 ijms-22-04116-f006:**
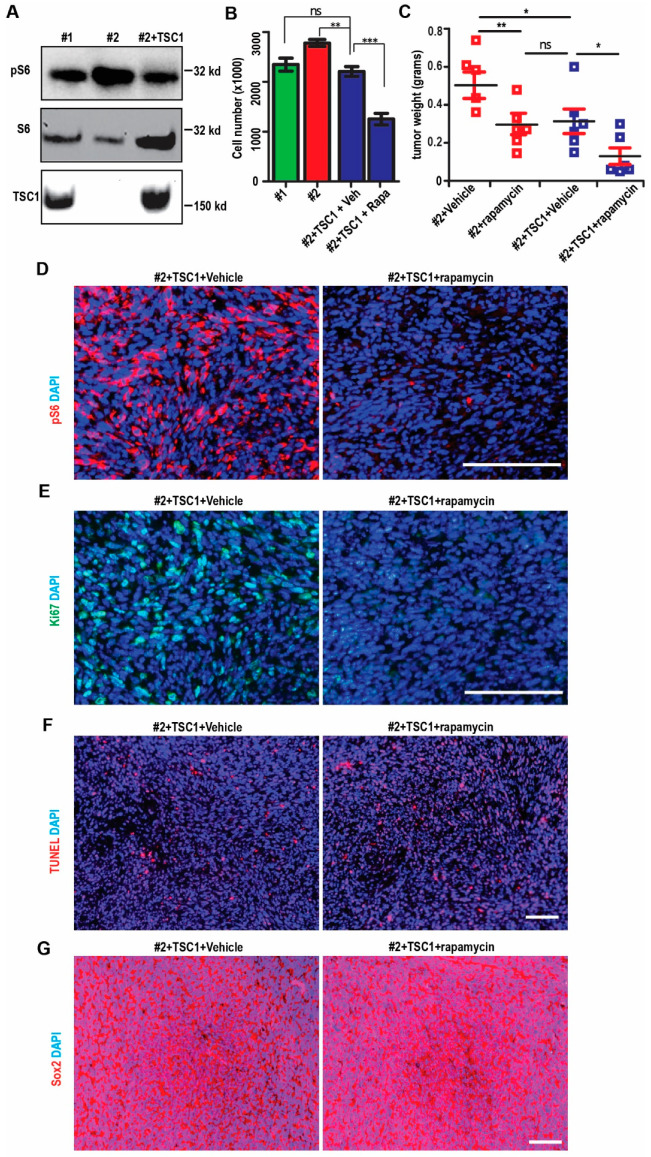
Re-expressing human TSC1 in Ast #2 cells and tumor. (**A**) Lysates extracted from Ast #1, Ast #2 infected with lentiviral vector or lentiviral hTSC1 were analyzed by Western blot using antibodies against TSC1, pS6RP, and total S6RP as indicated. Images were representative for 3 independent experiments. (**B**) Mean ± S.E. of cell number from Ast #1, Ast #2, Ast #2 + hTSC1 treated with vehicle or Rapamycin for 5 days in culture are shown. Experiments were repeated 6 times. (**C**) Mean ± S.E. of the tumor weight from Ast #2 and Ast #2 + hTSC1 treated with vehicle or Rapamycin for 2 weeks are shown. n = 5–6. (**D**–**G**) Immunofluorescent staining of pS6RP (**D**), Ki67 (**E**), TUNEL (**F**), Sox2 (**G**), and DAPI in tumors from Ast #2 + hTSC1 treated with vehicle or Rapamycin for 2 weeks. Two-way ANOVA and Tukey post hoc tests were used for statistic tests. ns: no significance; * *p* < 0.05, ** *p* < 0.01, *** *p* < 0.001. Bar = 50 µm.

## Data Availability

The data presented in this study are available on request from the corresponding author.

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
