# Peer review of "The Characterization of a Subependymal Giant Astrocytoma-Like Cell Line from Murine Astrocyte with mTORC1 Hyperactivation"

_ijms, 2021, doi:10.3390/ijms22084116_

Round 1

Reviewer 1 Report

Summary:

This purpose of this study was to reveal the mechanisms for tumorigenesis and progression of SEGA using TSC1-deficient neural cells. A created novel SEGA-like cell line that is invaluable for studying mTORC1-driven molecular and pathological alterations in neurologic tissue was used in this study. As a future perspective, these cells would develop novel therapeutic strategy for TSC patients with SEGA.

Major points.

In introduction, remove the “obstructive”, please. Because it is currently said the mechanism of hydrocephalus not as simple as the obstructive-communicating theory. Just simply use the term as # hydrocephalus”.

“as resection is challenging due to the deep intracranial location” is controversial. Some strongly recommend surgery and may guideline as well. I recommend not to mention this part as this phrase is irrelevant to this study.

Reading thru this paper, I understood what the authors wanted to say. The way to write paper is not appropriate.

In the abstract, the purpose of this study was “to reveal the mechanisms for tumorigenesis and progression of SEGA”. However, this was not appropriately shown . In my understand, what you did in this study was that the authors created the SEGA-like cell line using TSC1-deficient neural cells.

In that case, you may just say the fact that you created the novel cell line.

Results has many parts that should be written in the methods part.

Abstract and contents discrepancy as I above mentioned.

As I thought this was very interesting paper, please revise the way you wrote more simply and clearly.

Author Response

We thank the Reviewer for his/her very good comments. Please see attachment for our response.

Reviewer 2 Report

Tang et al have developed a novel, neurally derived cell model with TSC1 loss, which recapitulates several of the features of SEGAs found in patients with TSC. This is important for the field, as cell models better mimicking the tissue-specific features of TSC are lacking. The work is largely presented well, but some additional comparisons would be useful for the reader.

Major points:

  1. What is the proposed reason for the disconnect between higher Sox2 protein in Ast#1 and #2 compared to primary cells, but lower SOX2 mRNA in #1 and #2 compared to neurospheres from primary cells (Figure 1)?
  2. Figure 2: The authors state “Rapamycin treatment significantly decreased the proliferation in both cell lines for 5 days in culture (Figs. 2D and 2E)” – however no direct comparison is shown to untreated growth rates for these cells. Comparing by eye, the rapamycin treatment growth does not look much reduced when compared to Fig 2A 32 passage timepoint. Additionally, the statement “Interestingly, we noticed that Ast #2 cells proliferate more and have more Ki67+ cells after rapamycin treatment (Figs. 2D and 2E)” seems incomplete. It needs added that this is compared to Ast #1 cells with similar treatment (as opposed to increased proliferation following rapamycin treatment compared to untreated which it could otherwise be interpreted as).
  3. Figure 4 – it would have been nice to see whether Ast #2 are more tumorigenic than Ast #1, but this was not compared in the xenograft model. Does the downregulation of p53 seen in both #1 and #2 play a role in tumorigenesis or is it down to TSC1 alone?
  4. Figure 6: it would have been good to include #2 growth in Figure 6B for complete comparison. In Figure 6C – similarly, it would have been good to include the tumour weight of #2 tumours to those with TSC1 re-expression, to show the impact of mTORC1 inhibition by genetic means, not just chemical means.

Minor Points:

  1. There is an increase in Sox2 protein (Figure 1D), but no stats (densitometry) to really support the use of ‘significant’. Perhaps change to marked increase?
  2. Result section describing Figure 3 ends “These results suggested that the spontaneous immortalization of Ast #1 and Ast #2 cells might be through the mTORC1 independent mechanisms to inactivate p53.” This needs a reference.
  3. The discussion states “Although we are not pursuing a gene therapy in the current manuscript, we found that TSC1 replenishment along with Rapamycin decreased proliferation and growth in TSC1-deficient tumors, suggesting a more effective strategy for TSC patients with the combinational therapy” This would be easier for the reader to see if the tumour weights were directly compared (#2 v #2+Rapa v #2+TSC1 v #2+TSC1+rapa) as mentioned above for Figure 6.
  4. The authors should specify the statistic test used in each legend for clarity.

Author Response

(The authors gave the same response as above.)

Round 2

Reviewer 1 Report

Thank you very much for reflecting my comments. One last, for the hydrocephalus, the following reference may be helpful for your minor revision if you may think it is relevant.  

Experience using mTOR inhibitors for subependymal giant cell astrocytoma in tuberous sclerosis complex at a single facility.

Tomoto K, Fujimoto A, Inenaga C, Okanishi T, Imai S, Ogai M, Fukunaga A, Nakamura H, Sato K, Obana A, Masui T, Arai Y, Enoki H.BMC Neurol. 2021 Mar 31;21(1):139. doi: 10.1186/s12883-021-02160-5.PMID: 33784976

Author Response

Thank you very much for reflecting my comments. One last, for the hydrocephalus, the following reference may be helpful for your minor revision if you may think it is relevant. 

We thank the reviewer for the commnets. We include this helpful reference in our manuscript. Please check the newer version for the update.